# Fatty Acid Amides Synthesized from Andiroba Oil (*Carapa guianensis* Aublet.) Exhibit Anticonvulsant Action with Modulation on GABA-A Receptor in Mice: A Putative Therapeutic Option

**DOI:** 10.3390/ph13030043

**Published:** 2020-03-10

**Authors:** Fábio Rodrigues de Oliveira, Keuri Eleuterio Rodrigues, Moisés Hamoy, Ícaro Rodrigues Sarquis, Akira Otake Hamoy, Maria Elena Crespo Lopez, Irlon Maciel Ferreira, Barbarella de Matos Macchi, José Luiz Martins do Nascimento

**Affiliations:** 1Programa de Pós graduação em Neurociências e Biologia Celular, Instituto de Ciências Biológicas, Universidade Federal do Pará, Belém 66075-110, Brazil; oliveirafabio.fr@gmail.com (F.R.d.O.);; 2Laboratório de Neuroquímica Molecular e Celular, Instituto de Ciências Biológicas, Universidade Federal do Pará, Belém 66075-110, Brazil; 3Laboratório de Controle de Qualidade e Bromatologia, Curso de Farmácia, Departamento de Ciências Biológicas e da Saúde, Universidade Federal do Amapá, Macapá 68902-280, Brazil; 4Laboratório de Farmacologia e Toxicologia de Produtos Naturais, Instituto de Ciências Biológicas, Universidade Federal do Pará, Belém 66075-110, Brazil; 5Laboratório de Biocatálise e Síntese Orgânica Aplicada, Departamento de Ciências Exatas e Tecnológicas, Universidade Federal do Amapá, Macapá 68902-280, Brazil; 6Programa de Pós graduação em Ciências Farmacêuticas, Departamento de Ciências Biológicas e da saúde, Universidade Federal do Amapá, Macapá 68902-280, Brazil; 7Laboratório de Farmacologia Molecular, Instituto de Ciências Biológicas, Universidade Federal do Pará, Belém 66075-110, Brazil; 8Instituto Nacional de Ciência e Tecnologia em Neuroimunomodulação (INCT-NIM), Rio de Janeiro, RJ 21040-900, Brazil

**Keywords:** seizures, pentylenetetrazole, electrocorticographic recordings, biocatalysis, palmitoylethanolamide

## Abstract

Epilepsy is a chronic neurological disease characterized by excessive neuronal activity leading to seizure; about 30% of affected patients suffer from the refractory and pharmacoresistant form of the disease. The anticonvulsant drugs currently used for seizure control are associated with adverse reactions, making it important to search for more effective drugs with fewer adverse reactions. There is increasing evidence that endocannabinoids can pharmacologically modulate action against seizure and antiepileptic disorders. Therefore, the objective of this study is to investigate the anticonvulsant effects of fatty acid amides (FAAs) in a pentylenetetrazole (PTZ)-induced seizure model in mice. FAAs (FAA1 and FAA2) are obtained from *Carapa guianensis* oil by biocatalysis and are characterized by Fourier Transform Infrared Analysis (FT-IR) and Gas Chromatography-Mass Spectrometry (GC-MS). Only FAA1 is effective in controlling the increased latency time of the first myoclonic jerk and in significantly decreasing the total duration of tonic-clonic seizures relative to the pentylenetetrazol model. Also, electrocortical alterations produced by pentylenetetrazol are reduced when treated by FAA1 that subsequently decreased wave amplitude and energy in Beta rhythm. The anticonvulsant effects of FAA1 are reversed by flumazenil, a benzodiazepine antagonist on Gamma-Aminobutyric Acid-A (GABA-A) receptors, indicating a mode of action via the benzodiazepine site of these receptors. To conclude, the FAA obtained from *C. guianensis* oil is promising against PTZ-induced seizures.

## 1. Introduction

Epilepsy is one of the most common chronic neurological diseases, affecting about 50 million people worldwide of all ages, races, and both genders, of which about 80% live in developing countries [1,2]. This disease is characterized by recurrent excessive or synchronous neuronal activity leading to seizures [3].

These seizures are controlled by the chronic use of anticonvulsant drugs, which generate multiple adverse reactions, such as headache, fatigue, dizziness, sedation, and nausea, and in many cases, the occurrence of neurological problems such as anxiety, depression, or sleep disorders [4,5,6]. Nevertheless, even with a range of adverse effects, anticonvulsant drugs are widely used to control epilepsy, but about 30% of patients suffer from refractory and pharmacoresistant forms, presenting persistent and difficult to control seizures [2,7].

Benzodiazepine (BDZ) drugs, such as lorazepam and diazepam, have been shown to possess effective anticonvulsant activities and enhance GABA-mediated inhibition [8]. These drugs are effective in controlling seizures by producing allosteric changes in GABA-A receptors, increasing GABAergic neurotransmission, and decreasing neuronal excitability. However, like other anticonvulsant drugs, they are associated with a range of adverse reactions, such as myorelaxation and sedation [9]. Thus, it is necessary to search for new drugs, effective and with less adverse reactions, to treat seizures and epilepsy.

To this end, endocannabinoids, such as anandamide and 2-arachidonoyl-glycerol (2-AG) and their analogs, palmitoyethanolamine (PEA) and N-oleyethanolamine (OEA), which are known as fatty acid amides (FAA) acting as endogenous lipid signaling molecule analogues, have been extensively studied as a target for new therapeutic options for central nervous system (CNS) disorders [10,11,12,13,14], including the prevention of epileptic seizures [15,16,17,18,19,20]. Endocannabinoids, natural or synthetic, have attracted the attention of researchers due to their various biological actions, including neuroprotective and antiepileptic, with effective anticonvulsant properties [21,22,23,24,25,26,27,28,29,30,31].

The most frequent biosynthetic routes for these endocannabinoids are through arachidonic acid and involve compounds that are easily obtained (by chemical synthesis) and can be developed in the presence of a metallic catalyst [32,33] or biocatalyst [34,35]. Another possibility is based on the direct reaction from rich fatty acid oils from plants with amine that appear to be feasible for a biosynthetic route [36,37,38].

Several plant genera have been identified as good sources of fixed oils, especially *Carapa guianensis* Aublet, an Amazonian species popularly known as andiroba, rich in fatty acids such as oleic, palmitic, myristic, linoleic, stearic, and arachidic [39,40,41,42,43,44]. Added to its fatty acid-rich composition, andiroba oil is easy to obtain and contains substances that can be used as precursors for the synthesis of endocannabinoids. This species is widely used in traditional medicine to treat seizures [45,46].

Thus, we hypothesize that andiroba oil can now be employed for the construction of the amide bond and provides a good source of seizure-effective endocannabinoid-like FAA.

Therefore, the goal of this study is to evaluate the anticonvulsant potential of FAA obtained by biocatalysis technique in *C. guianensis* oil in pentylenetetrazole (PTZ) -induced seizures in mice.

## 2. Results

### 2.1. Fourier Transform Infrared Analysis (FT-IR) and Gas Chromatography-Mass Spectrometry (GC-MS)

Both samples from Andiroba oils (AO1 and AO2) and fatty acid amides (FAA1 and FAA2) showed a similar Fourier transform infrared analysis (FT-IR) profile (Figure 1). The main signals were at 2915–2858 cm^−1^ and included asymmetric CH_2_ and CH_3_ vibrations of the fatty acid chains of andiroba oil, as well as fatty amides. The spectrum for oil showed a peak identified at 1160 cm^−1^, characteristic of the C - O group, and for the peak for the C = O carbonyl group at 1750 cm^−1^, which were due to the glycerol group present in the triglyceride of the oils, whereas for amides this was due to stretching (C = O) with a slight signal shift at 1657 cm^−1^. Another characteristic peak of synthesized amides was observed at 1525 cm^−1^ due to stretching of the C-H bond at 1520 cm^−1^, while the N-H bond was stretching to the 3299 cm^−1^ peak.

Interestingly, after treatment of AO1 and AO2 with ethanolamine, all fatty acids were converted to fatty acid amides, since the structural features of the analyzed molecules were identified by gas chromatography mass spectrometry (GC-MS) and deduced from the fragmentation profile by mass spectrometry (MS), characteristic of fatty ethanolamides resulting from a McLafferty rearrangement and β-cleavage forming fragmentation ions in 103 and 116 *m*/*z*, respectively [47]. These fragments can be used for structure elucidation and are represented in Table 1.

### 2.2. Effect of Andiroba Oil and Fatty Acid Amides on pentylenetetrazole (PTZ)-Induced Seizure Behavior

FAA1 significantly increased the latency time of the first myoclonic spasm (86.57 ± 24.01 s) compared to the vehicle + PTZ group (53.00 ± 9.01 s) and other groups (FAA2, AO1, and AO2) (*p <* 0.001; *F* = 1747) (Figure 2). The FAA2, AO1, and AO2 groups showed no statistical difference in relation to the vehicle + PTZ group (58.71 ± 8.92, 51.25 ± 12.42, and 62.13 ± 20.78 s, respectively).

The time-of-onset of tonic-clonic seizures was extended by the action of FAA1 (349.4 ± 148.9 s) with the statistical difference in relation to the vehicle + PTZ group (137.0 ± 83.02 s) and all other groups (AO1 97.75 ± 51.49 s, FAA2 159.8 ± 105.8 s, and AO2 147.4 ± 123.5 s) (*p* < 0.0001; *F* = 28.03).

The mean duration of tonic-clonic seizures induced for vehicle + PTZ was 22.00 ± 7.32 s, compared to 10.29 ± 2.69 s for the FAA1 group, which was the only group in which this parameter was significantly reduced (*p* < 0.0001; *F* = 13.60). The other groups. FAA2, AO1, and AO2 (16.43 ± 10.50 s, 16.43 ± 5.47 s, and 16.43 ± 5.06 s respectively), showed no statistically significant difference.

Concerning all evaluated parameters, BDZ presented statistical differences in relation to vehicle + PTZ and the other evaluated groups, showing its efficacy as a standard drug used in this experimental model.

Added to the improvement of all assessed seizure behavioral parameters, only the FAA1-treated group was selected for electrocorticographic evaluation.

### 2.3. Electrocorticographic Analysis of Amides Synthesized from Andiroba Oil

Representative data for Control, FAA1, BDZ, flumazenil (FMZ) and Vehicle + PTZ groups are shown in Figure 3. The electrocorticographic tracing (ECoG) measured the voltage as a function of time, giving rise to a time series in which the Vehicle + PTZ group presented an increase in the amplitude of the recorded waves, with characteristic peaks of generalized seizures relative to the control group. The other groups showed no variation in the amplitude of the recorded waves when compared to the control.

Considering the time series analysis, we performed the Fourier transform (FFT), converting the time series into a frequency graph, thereby obtaining a power spectrum which showed the average power in each frequency range. Concerning the control group, there was a frequency distribution ranging from 0 to 40 Hz, with energy concentrated in the frequency range that went up to 5 Hz, thus having the predominance of the lower power spectrum. Regarding the Vehicle + PTZ group, the power spectrum changed, increasing its energy over the various frequency ranges when compared to the control group. Viewing the FAA1, FMZ and BDZ groups, the frequency spectrograms were similar to that expressed by the control group (Figure 4A).

The mean power of the obtained baseline traces FMZ, BDZ and FAA1 groups (0.283 ± 0.101, 0.313 ± 0.145 and 0.392 ± 0.187 mV^2^/Hz × 10^−3^, respectively) with no statistical difference compared to the control (0.337 ± 0.127 mV^2^/Hz × 10^−3^) (Figure 4B). The Vehicle + PTZ group presented a mean power of 10.63 ± 4.191 mV^2^/Hz × 10^−3^, being statistically different in relation to the other groups (*p* < 0.0001; *F* = 53.96).

As only the vehicle + PTZ group showed a statistical difference relative to the control, we compared the brainwave pattern (Delta, Theta, Alpha, Beta, and Gamma) in these two groups. We detected a change in brain rhythms (Figure 5). The highlight was the power increase in the beta frequency range in vehicle + PTZ with an average of 8.537 ± 2.250 mV^2^ / Hz × 10^−3^ compared to the control with a mean of 0.029 ± 0.017 mV^2^/Hz × 10^−3^. Thus, the beta frequency band, which is typically associated with the activation of the motor system, was selected as the interest for the assessment of the effect of FAA1 on the PTZ-induced seizure.

### 2.4. PTZ-Enhanced Brain Activity and Beta Oscillation is Reduced by FAA1

Electrocorticographic recordings were performed to determine the effect of treatments on seizures induced by PTZ. The results showed smaller amplitude variations for BDZ and FAA1, demonstrating a reduction in seizure outbreak when compared to the Vehicle + PTZ record (Figure 6).

During the mean power evaluation, the FAA1 group (3.76 ± 1.85 mV^2^/Hz × 10^−3^) showed a statistical difference (Figure 7A) with the control and vehicle + PTZ (0.33 ± 0.12 and 10.63 ± 4.19 mV^2^/Hz × 10^−3^, respectively) (*p* < 0.0001; *F* = 28.61), but the same did not occur with BDZ (1.40 ± 0.69 mV^2^/Hz × 10^−3^).

FAA1 reduced beta oscillation compared to Vehicle + PTZ with values of 3.99 ± 2.17 and 8.53 ± 2.25 mV^2^/Hz × 10^−3^, respectively (*p* < 0.0001; *F* = 43.08). However, it significantly maintained higher energy compared to the control (0.02 ± 0.01 mV^2^/Hz × 10^−3^) and BDZ (1.00 ± 0.51 mV^2^/Hz × 10^−3^) (Figure 7B).

### 2.5. Flumazenil Reverses the Anticonvulsant Effect of FAA1

FAA1 was effective in controlling triggered seizures by PTZ and, in an attempt to elucidate the possible mechanism of action of FAA1, Flumazenil (a benzodiazepine antagonist that will act on GABA-A receptors) was used. After its administration, the return to seizure was observed in groups BDZ and FAA1. Electrocorticographic tracings showed increased wave amplitudes in these groups (Figure 8), with peaks that mimicked epileptic seizures. During the spectrogram, all groups showed an increased power spectrum, which was similar to the PTZ spectrum.

After Flumazenil, the FAA1 and BDZ groups showed increased beta-force energy with means of 6.68 ± 2.00 and 8.52 ± 1.17 mV2/Hz × 10^−3^, respectively, with no statistical difference compared to the vehicle + PTZ group (8.53 ± 2.25 mV2/Hz × 10^−3^) (Figure 9).

## 3. Discussion

An important finding in the present study was demonstrating that fatty acid amides (FAAs) synthesized from andiroba oil (*Carapa guianensis*) were able to decrease seizures evoked by pentylenetetrazole (PTZ) with potential mechanisms involved in neuroprotection. These effects were reversed by flumazenil, a benzodiazepine antagonist on Gamma-aminobutyric acid (GABA)-A receptors. Additionally, we show that andiroba oil (AO) collected and distributed from the Amazon is a good source for fatty acid amide production and can be employed similarly to anticonvulsant drugs in a pentylenetetrazole model of seizure in mice.

One of the proposed mechanisms for seizure neuroprotection is the interaction between endocannabinoid compounds and GABAergic systems. These interactions have been previously demonstrated (e.g., cannabinoid receptor agonists were effective in protecting against pentylenetetrazole-induced seizures) [48]. Currently, PTZ is considered an important model of myoclonic and generalized tonic-clonic seizures for screening therapeutic advances for anticonvulsant compounds [49,50].

Oils undergo the biocatalysis process using ethanolamide, as evidenced in the Mass spectrometry (MS) profile when samples showed characteristic signals ofFAAs, as observed in other characterization studies of these compounds using Fourier transform infrared analysis (FTIR), and in previous studies that have used the Gas chromatography mass spectrometry (GC-MS) technique to quantify fatty acid and fatty acid amide content in various sample types [47,51]. Other studies also have obtained fatty amides from direct amidation of triglycerides from other natural sources [38,52].

Andiroba oil is very common in the Amazon territory and is characterized by its high fatty acid content and good extraction yield (over 20% according to the form of extraction used) [46]. These characteristics make this a cheap, accessible source and a good substrate candidate for the production of FAAs.

Our GC-MS results showed a variation in the number of FAAs produced. This variation is attributed to the fatty acid content present in each oil, which might be related to seasonal changes [53,54]. The amount of palmitoyethanolamine (PEA) (26.5%) and stearoyl ethanolamide (11.1%) found in FAA1 compared to FAA2 (8.5% and 5.5%, respectively) confirm this difference. Another study also evaluated fatty acid composition in andiroba oil and used the enzymatic catalytic process with lipases from *Candida antarctica* -B (CAL-B) and ethanol for biodiesel production, obtaining a yield of over 90% of fatty acid ethyl esters, with oleic and palmitic acid as major constituents [44].

This difference was reflected in the biological activity presented by FAA1 and FAA2, and only pre-treated mice with FAA1 produced significant changes in myoclonic, tonic-clonic latency, and duration of seizures in the PTZ-induced model, probably due to its higher composition in PEA. PEA is a lipid modulator recognized as a naturally occurring amide of ethanolamide and palmitic acid [55]. Related studies show that PEA exerts neuroprotective effects and was able to attenuate effects in the electroshock-induced seizure model of mice [12,56,57,58]. Moreover, other compounds structurally related to PEA mimicked several endocannabinoid-driven actions, including anticonvulsant activity [28,59,60,61]. These data are consistent with the results reported here.

Interestingly, our results show that AO was unable to affect behavioral seizures induced by PTZ in its original composition, as well as in other studies with some fatty acids or analogs, such as palmitic acid, hexadecanol, and 16-hydroxypalmitic acid, and oleic acid; even when used at high doses it showed no significant protective effects, either [12,62].

Behavioral assessment and electrical recordings are extremely helpful for comparing alterations produced by neuronal excitability that triggers seizures and epileptic conditions [49,50,63]. Our data with FAA1 revealed that on behavioral evaluation, the increased latency and seizure threshold was confirmed by eletrocorticographic (ECoG) recording, since the observation of behavior could not be quantified. These recordings showed a decrease in the spike and energy of waves. Thus, FAA1 produces effective protection in the control of seizures induced by PTZ.

Seen in a previous study, Smythe et al., [64] evaluated the electrocorticogram of rat pups and demonstrated that PTZ-treated animals had continuous shooting and high amplitude waves, in which this effect was reversed by BDZ. Our results using the same drugs corroborate this idea, besides showing that FAA1 treatment reduced the amplitude of the waves, as well as the characteristic peak bursts of seizures. Additionally, a power reduction in the beta frequency band occurs.

Increased beta strength is related to diseases involving motor impairment, such as Parkinson’s disease [65]. Our results show that FAA1 prevented the energy increase in beta frequency in the model of the seizures.

One of the proposed mechanisms for seizure neuroprotection is the interaction between endocannabinoid compounds and GABAergic systems, as already demonstrated with cannabinoid receptor agonists effective in protecting against pentylenetetrazole-induced seizures [48]. GABA is the major inhibitory neurotransmitter in the brain, and its inhibition seems to be one of the underlying factors in epilepsy [66,67].

PTZ, acting as GABA-A receptor antagonists, is a selective blocker of the chloride channel [68,69]. To investigate the mechanism through which FAA1 was responsible for reducing seizures, flumazenil, a benzodiazepine antagonist, was used to observe the behavior of ECoG. After the administration of this drug, all groups demonstrated resumption of seizures and beta oscillation patterns similar to the PTZ group.

Therefore, FAA1 elicits potentiation via the benzodiazepinic site against PTZ- induced seizures. Additionally, as expected, flumazenil application had a consistent inhibitory action and could reverse these actions. These results provide evidence that FAA1 is involved in the regulation of GABA-A receptors.

Other amides, such as Oleamide, a primary amide, also have GABA-A receptor-related endocannabinoid activity, such as Anandamide and 2-arachidonoyl-glycerol (2-AG), that are released and able to reverse the convulsant effects produced by PTZ [70,71,72,73]. The 2-AG synthesis is reduced during the ictal phase of the epileptic process in patients with temporal lobe epilepsy, and this reduction might be due to impaired expression of Diacil glycerol lipase and degradation enzymes (Monoacylglycerol lipase and α/β-hydrolase) that undergo alteration leading to a decreased release of GABA [74,75]. Therefore, due to the mechanism demonstrated by FAA1, its supplementation may help to improve GABAergic signaling.

## 4. Materials and Methods

### 4.1. Chemicals

Ketamine hydrochloride was purchased from König (Santana de Parnaíba, SP, Brazil), xylazine from Vallée (Montes Claros, MG, Brazil), lidocaine from Hipolabor (Sabará, MG, Brazil), pentylenetetrazole from Sigma-Aldrich (St Louis, MO, USA), diazepam from União Quimica (Embu-Guaçu, SP, Brazil), and flumazenil (Lenazen^®^-Teuto). Ethanolamine (99%) and Lipase B *Candida antarctica* immobilized (CAL-B) (>2000 U/g) were purchased from Sigma-Aldrich (São Paulo, SP, Brazil). *n*-Hexane and ethyl acetate (PA grade) were obtained from Quimis (São Paulo, SP, Brazil) and used without further purification.

### 4.2. Plant Material and Oil Extraction of Carapa Guianensis

Botanical material was collected in Mazagão city-Amapá, Brazil (00°13′48.3″ S 51°24′37.4″ W), identified by a botanist (Dr. Rosângela Sarquis) and a voucher specimen deposited in the herbarium of the Brazilian Agricultural Research Company (EMBRAPA) under number N°194102.

The seeds were collected in two periods, according to Amazon seasonality [76,77]. The first collection was performed on the tenth day of March (AO1) and the second on the tenth day of the month of June (AO2) in the same place. The Andiroba oils (AO1 and AO2) were obtained by the traditional method of cooking the seeds and resting for 72 h. Then, the almonds were removed from shells, and the fruit mass was prepared and, finally, the oil collected [46].

### 4.3. Process for Obtaining Fat Acid Amides by Lipases from Candida antarctica -B (CAL-B)

The enzymatic amidation reaction was performed separately for Andiroba oils (AO1 and AO2), generating their respective fatty acid amides (FAA1 and FAA2), following the methodology described by Araújo et al. [37]. Briefly, andiroba oil (3 mL), ethanolamine (9 mL), and CAL-B (150 mg) were added to an Erlenmeyer flask (25 mL). The mixture was stirred for 24 h under controlled conditions (150 rpm at 35 °C), then filtered for enzyme retention and washed with 15 mL chloroform. Next, 15 mL of distilled water was added and the organic phase extracted with chloroform (2 × 15 mL) and anhydrous sodium sulfate and again filtered. Finally, the reaction was purified by column chromatography on silica gel 60 using an hexane:ethyl acetate (9:1) mixture. The reaction yield was calculated by the gravimetric method. The obtained amides (FAA1 and FAA2) were characterized by Infrared-Fourier Transform Spectroscopy and Gas Chromatography-Mass Spectrometry (CG-MS).

### 4.4. Fourier Transform Infrared Analysis (FT-IR)

FT-IR spectra were recorded on a Shimadzu IR Affinity spectrometer (Oils and FAAs), and samples were prepared as thin films on KBr disks. The transmittance was expressed in cm^−1^ of the band between 4000 and 400 cm^−1^ with the resolution of 4 cm^−1^ with 64 scans.

### 4.5. Gas Chromatography-Mass Spectrometry (GC-MS)

The analysis from FAAs was conducted using a Gas Chromatograph (GCMS-QP 2010) equipped with an auto-sampler injection AOC-20i (Shimadzu, Osaka, Japan). Electron impact detection was used as a detector (Shimadzu MS2010 Plus), with the electronic impact of 70 eV and fragments detected from 50 to 500 Da. Separations were performed on a fused silica capillary column (RTX-5MS with i.d. = 0.25 mm, length = 30 m, and film thickness = 0.25 µm) in a stream of helium 1.0 mL/min. The sample was solubilized in ethyl acetate (3 µg/mL), and 1.0 µL of the solution was subjected to the following experimental conditions: injector temperature, 210 °C; detector temperature, 250 °C; carrier gas, Helium; flow rate 1.0 mL/min; split injection with split ratio 1/15. The column temperature was programmed from 130 °C, with an increase of 5 °C/min to 290 °C, ending with a 5 min isothermal at this temperature, the total analysis time was 39 min.

### 4.6. Animals

Adult Swiss mice (20–26 g) were obtained from the Central Animal Facility of the Federal University of Pará. All animals were maintained in a controlled environment (23–25 °C and 12 h light-dark cycle), with food and water available ad libitum. All experimental procedures were carried out in accordance with the principles of laboratory animal care and were approved by the Animal Ethics Committee of the Federal University of Pará (CEUA N°4343290519). A total number of 75 animals were used in the present study (30 for behavioral seizure analyses after treatment with andiroba oils and fatty acid amides and 45 for electrocorticographic recordings after treatment with fatty acid amides). Additionally, all procedures were conducted to prevent animal suffering and distress.

### 4.7. Experimental Procedures

Regarding administration to animals, both AO and FAA solutions were prepared (60 mg/mL) using a mixture of dimethyl sulfoxide and saline 0.09% (1:20 *v*/*v*) [12]. Concerning behavioral seizure analyses, six different animal groups (*n* = 5 mice per group) were used. Group 1: Vehicle group (received a dimethyl sulfoxide and saline 0.09% solution 1:20 *v*/*v*). Groups 2, 3, 4, and 5 received a single dose at 300 mg/kg of AO1, AO2, FAA1, and FAA2, respectively. Group 6 received a single dose of BDZ at 5 mg/kg. After 30 min, every group was treated with PTZ at 60 mg/kg. All drugs were administered intraperitoneally, not exceeding 200 µL per animal.

The determination of the FAA dose was based on previous literature [78] and a dose-response curve using linear regression based on the increase in latency of tonic-clonic seizures. The determination of the PTZ dose has been reported before [79].

Regarding electrocorticographic recordings, animal groups (*n* = 9) were used and pretreated with vehicle, FAA1 (300 mg/kg), and BDZ (5 mg/kg) before PTZ administration, and after recording time, a flumazenil (FMZ) injection (0.1 mg/kg i.p.) was administered. Added to the treated groups, an FMZ and a control group (saline treated) were used to record basal parameters.

Concerning both behavioral seizures and electrocorticographic analysis, the evaluated parameters and recordings were scored by an investigator that was blinded to the animal group and treatment conditions.

### 4.8. Behavioral Seizure Analyses

After PTZ injection, animals were monitored for 10 min to the latency time of the first myoclonic spasm and first generalized tonic-clonic seizure, as well as the total duration of tonic-clonic seizures [49,79].

### 4.9. Surgery for Electrode Placement

Animals were anesthetized with ketamine (100 mg/kg, i.p.) and xylazine hydrochloride (10 mg/kg, i.p.). After surgical procedures to expose the skull, two bilateral holes were drilled in the skull of the mouse using a dental drill. Stainless steel electrodes (tip exposure, 1.0 mm diameter) were placed on the dura mater above the frontal cortex at coordinates 1 mm posterior to the bregma ± 1.0 mm lateral-lateral (both hemispheres) [79,80]. A screw was fixed in the occipital skull region, and the electrodes were fixed with dental acrylic cement (Methyl methacrylate monomer). The whole experiment was performed in Faraday cages and the ground electrode was fixed on the animal’s right paw. The recording electrode was located on the right side of the hemisphere, and the electrode on the left side was used as a reference.

### 4.10. Eletrocorticographic Records

Five days after surgery, the electrodes were connected to a digital data-acquisition system composed by a high impedance amplifier (Grass Technologies, P511, Carlow, Ireland), an oscilloscope (Protek, 6510, Kaohsiung, Taiwan), and a board for data acquisition and digitalization (National Instruments, Austin, TX, USA). Data were continuously sampled at 1 kHz at a low pass of 3 kHz and a high pass of 0.3 Hz.

The recordings followed a standard protocol: 10 min of accommodation in carefully immobilized animals to avoid record interference. Data collection consisted of the initial baseline of the control groups and treated groups (as described above) for 10 min. After 20 min of treatment, PTZ was administered, and electrocorticographic activity was further recorded for 10 min. Afterward, Flumazenil was administered and recording was performed for the same time [50].

### 4.11. Data Analyses

Offline analysis was performed through a tool built using Python programming language (version 2.7). “NumPy” and “SciPy” libraries were used for the mathematical processing, and the “matplolib” library was used to obtain graphs and plots. A graphic interface was developed using the PyQt4 library. Spectrograms were calculated using a Hamming window with 256 points (256/1000 s). Regarding power spectral density (PSD), each frame was generated with an overlap of 128 points per window. Concerning each frame, the PSD was calculated by Welch’s average periodogram method. Frequency histograms were obtained by calculating the PSD of the signal using the Hamming window with 256 points without overlap, yielding a resolution of 1 Hz per bin. Each wave displayed in PSD was an average from a set of experiments. PSDs were calculated in each group, and the means were shown by individual bins. Analyses were performed in frequencies of up to 40 Hz, and split in bands according to Jalilifar, Yadollahpour, Moazedi, and Ghotbeddin (2017) in Delta (0.5–3.5 Hz), Theta (4–7.5 Hz), Alpha (8–12 Hz), Beta (13–20 Hz), and Gamma (20–40 Hz) for interpretation of the dynamics during seizure development [81].

### 4.12. Statistical Analyses

Results were expressed as mean ± standard deviation (SD). Normality and homogeneity of variances were verified using Kolmogorov–Smirnov and Levene’s tests, respectively. Since residuals were normally distributed, comparisons among the control, vehicle + PTZ, and other groups in seizure behavioral analyses (latencies and duration of seizures) and electrocorticographic results were performed using one-way Analysis of variance (ANOVA) and a Tukey post-test. A Student’s *t* test used for comparison between brain oscillation power of the control group and vehicle + PTZ group. The minimum significance level was set at *p* < 0.05 in all cases. The GraphPad^®^ Prism 6 software was used for all analyses.

## 5. Conclusions

To conclude, the biocatalysis process using ethanolamine from Andiroba oil was effective in the production of FAAs, and these were important modulators in controlling PTZ-induced seizures acting on the GABA-A receptor. Therefore, fatty acid amides from plants likely provide an important resource for the future discovery of new protective pharmaceutical agents in epilepsy.

## Figures and Tables

**Figure 1 pharmaceuticals-13-00043-f001:**
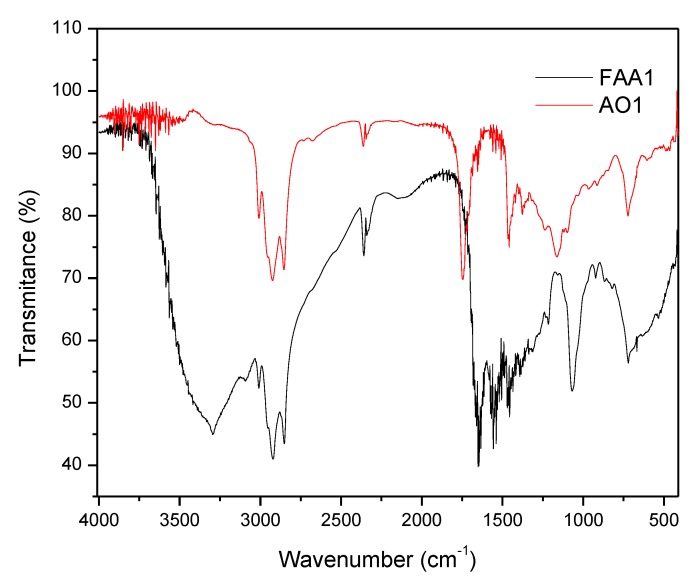
Fourier transform infrared analysis: FT-IR spectra of the sample from andiroba oil: AO 1 (red line) and fatty acid amides: FAA1 (black line).

**Figure 2 pharmaceuticals-13-00043-f002:**
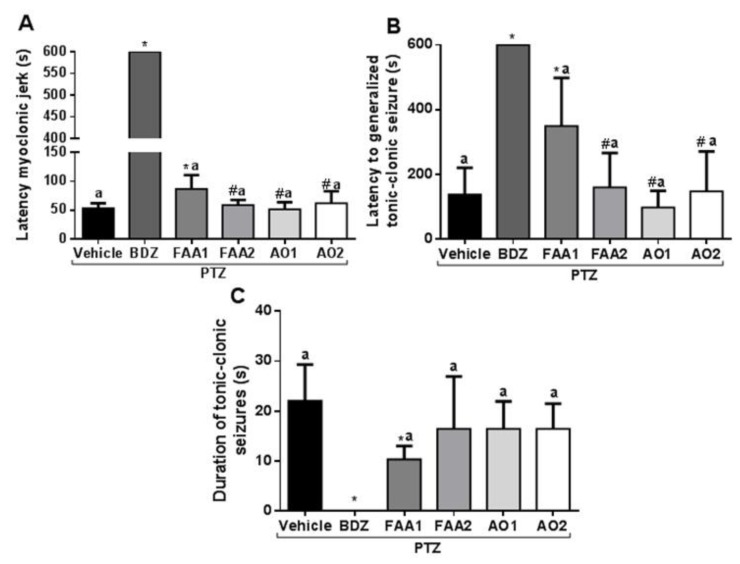
Effects of andiroba oils (AOs = 300 mg/kg) and fatty acid amides (FAAs = 300 mg/kg) on behavioral seizure parameters induced by PTZ (60 mg/kg). (**A**) The latency of the first myoclonic spasm, (**B**) the latency of the tonic-clonic spasm, and (**C**) duration of the tonic-clonic seizures. All results were expressed in seconds (mean ± SD). (*) Statistical difference with vehicle group; (#) difference with FAA1 and (a) difference with benzodiazepine (BDZ) group. Using one-way Analysis of variance-ANOVA, followed by a Tukey test (*p <* 0.05).

**Figure 3 pharmaceuticals-13-00043-f003:**
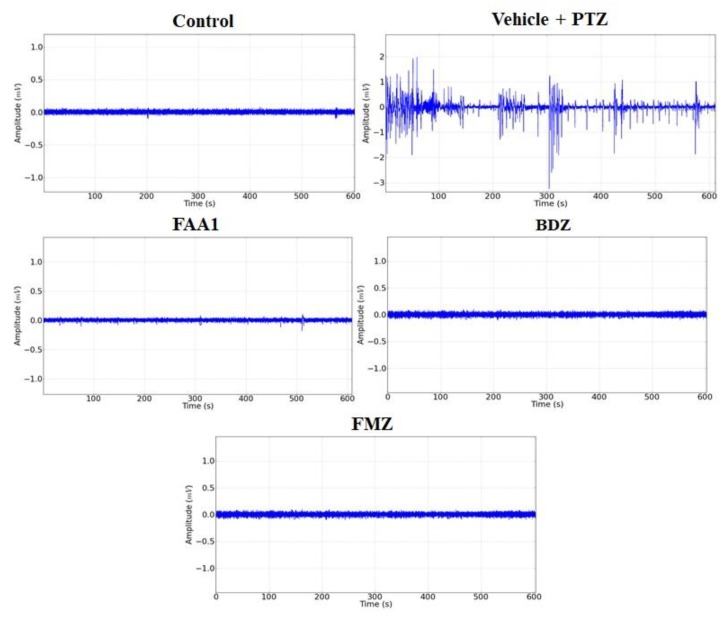
Representative data of electrocorticographic tracing for control, vehicle + PTZ (60 mg/kg), FAA1 (300 mg/kg), BDZ (5 mg/kg) and FMZ (0.1 mg/kg), lasting 600 s.

**Figure 4 pharmaceuticals-13-00043-f004:**
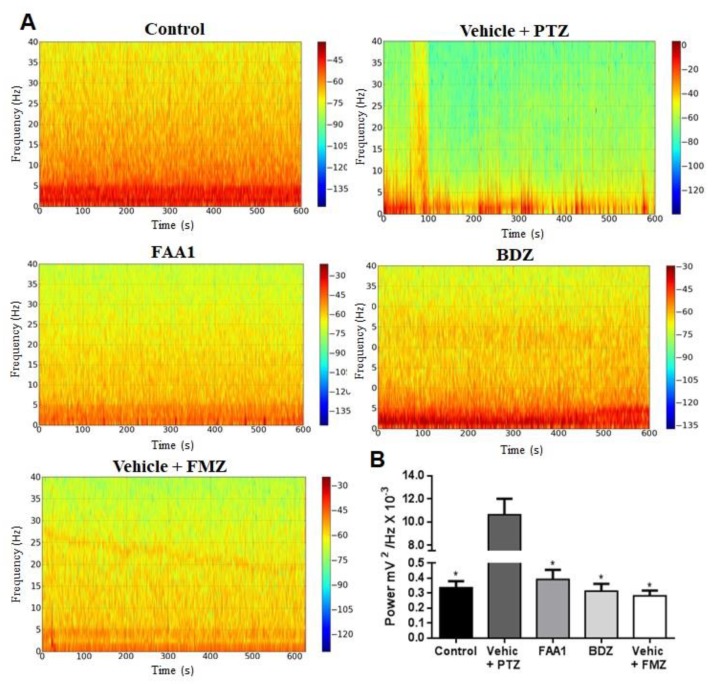
(**A**) Spectrogram of frequency distribution of control and treated groups with BDZ (5 mg/kg) FAA1 (300 mg/kg), FMZ (0.1 mg/kg) and Vehicle + PTZ (60 mg/kg) for 600 s. (**B**) Mean total power distribution. (*) Statistical difference with Vehicle + PTZ group (*p* < 0.0001, *n* = 9), using one-way ANOVA, followed by a Tukey test.

**Figure 5 pharmaceuticals-13-00043-f005:**
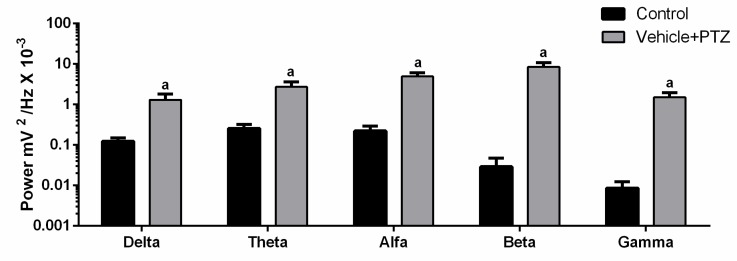
Comparison between brain oscillations power (Delta: 0.5–3 Hz, Theta: 3.5–7.5 Hz, Alpha: 8–12 Hz, Beta: 13–20 Hz and Gamma: 20–40 Hz) of the control group and vehicle + PTZ group (60 mg/kg). Graph expressed in logarithmic scale and (a) significant differences with control of each oscillation after Student’s T-test (*p* < 0.05, *n* = 9).

**Figure 6 pharmaceuticals-13-00043-f006:**
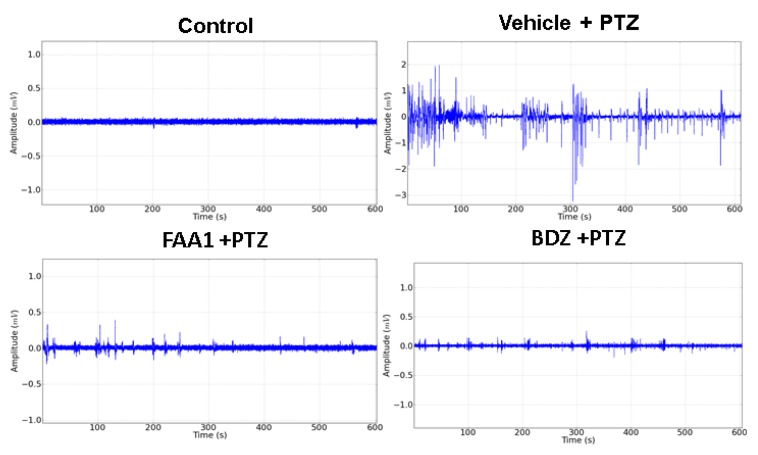
Electrocorticographic tracing of animals in the control group compared to the other groups treated with the vehicle, FAA1 (300 mg/kg), and BDZ (5 mg/kg), after PTZ (60 mg/kg) administration for 600 s.

**Figure 7 pharmaceuticals-13-00043-f007:**
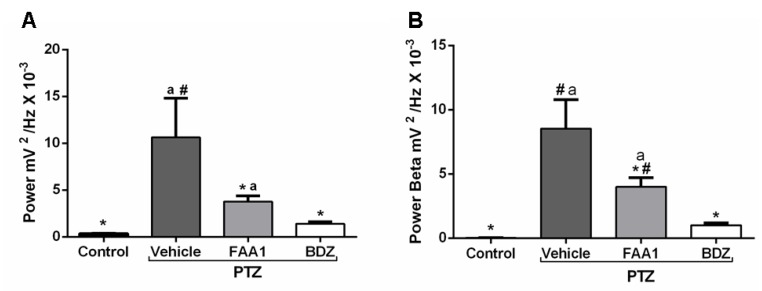
Mean total power records (**A**) and Beta oscillations (**B**) of control, vehicle, BDZ, and FAA1 mice groups after PTZ (60 mg/kg) administration. Results in mV^2^/Hz × 10^−3^ and represent mean ± SD (*n* = 9). Statistical difference in relation to the vehicle group (*) (*p* < 0.0001; *F* = 28.61); BDZ (#) (*p* < 0.05) and control group (a) (*p* < 0.05), using one-way ANOVA, followed by a Tukey test.

**Figure 8 pharmaceuticals-13-00043-f008:**
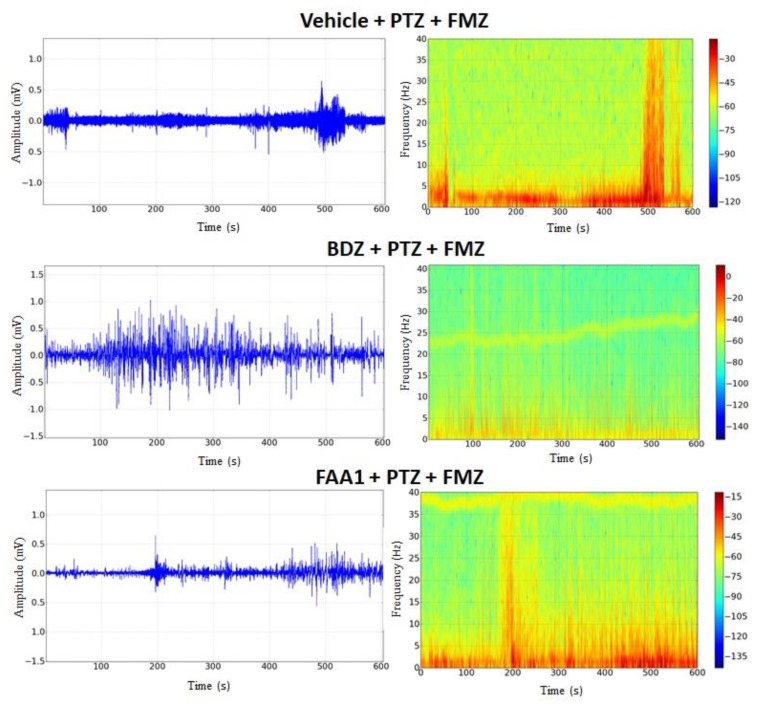
Electrocorticographic tracing and spectrogram recordings after flumazenil administration in control (FMZ) and treated groups with BDZ (5 mg/kg) and FAA1 (300 mg/kg), 10 min after seizure induction with PTZ (60 mg/kg). Recordings made for 600 s.

**Figure 9 pharmaceuticals-13-00043-f009:**
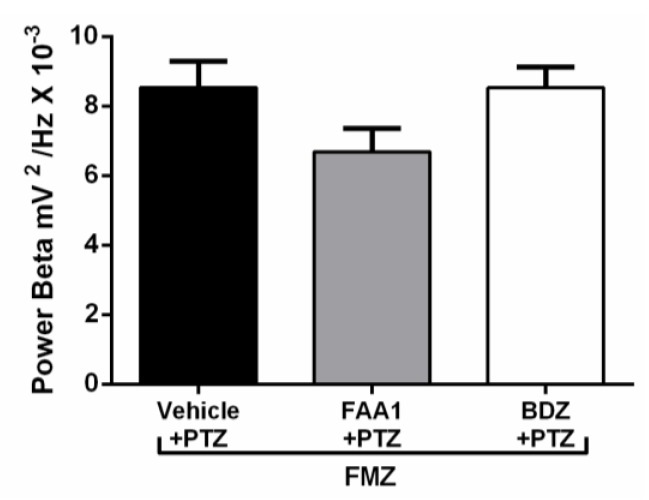
Means of power recordings in beta oscillations of groups treated with Vehicle, BDZ (5 mg/kg) and FAA1 (300 mg/kg) after induction of seizures with PTZ and subsequent administration of Flumazenil (0.1 mg/kg). Results in mV^2^/Hz × 10^−3^ and represent mean ± SD (*n* = 9), using one-way ANOVA, followed by a post-hoc Tukey test.

**Table 1 pharmaceuticals-13-00043-t001:** Composition of fatty acid amides (FAA) from andiroba oils (AO) by gas chromatography-mass spectrometry (GC-MS) catalyzed by lipases from *Candida antarctica* -B (CAL-B).

Peak	FAA	FAA1 ^a^ (%)	FAA2 ^a^ (%)	Values *m*/*z* (%)
1	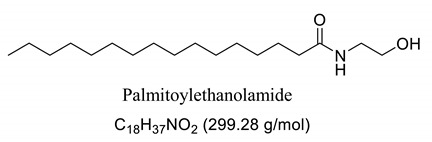	26.5	8.5	85 (100), 98 (48), 103 (26), 116 (10), 140 (12), 256 (10), 299 (2).
2	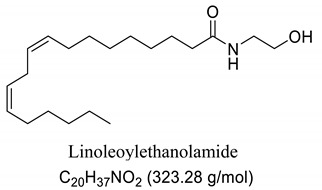	4.2	30.2	85 (65), 98 (100), 103 (56), 116 (45), 194 (12), 264 (10), 323 (2).
3	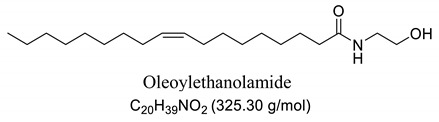	52.6	50.1	85 (65), 98 (100), 103 (50), 116 (50), 194 (12), 208 (18), 264 (10), 325 (5).
4	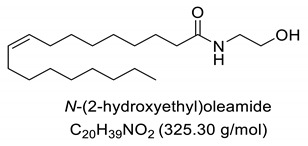	0.99	0.40	85 (75), 98 (100), 116 (23), 154 (24), 208 (20), 222 (18), 250 (16), 264 (14), 325 (2).
5	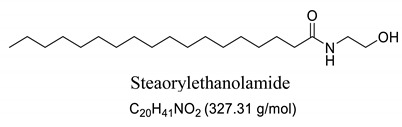	11.1	5.5	85 (100), 98 (70), 103 (35), 116 (12), 154 (13), 252 (14), 284 (8), 327 (2).
6	Not identified	4.56	5.3	--

^a^ Values were determined by integrating the areas of the compounds identified in the chromatogram.

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
