# Peer review of "Fatty Acid Amides Synthesized from Andiroba Oil (Carapa guianensis Aublet.) Exhibit Anticonvulsant Action with Modulation on GABA-A Receptor in Mice: A Putative Therapeutic Option"

_pharmaceuticals, 2020, doi:10.3390/ph13030043_

Round 1

Reviewer 1 Report

               In the manuscript entitled "Fatty Acid Amides Synthesized from Andiroba Oil (Carapa guianensis Aublet.) exhibit Anticonvulsant Action with Modulation on GABA-A Receptor in Mice: A Putative Therapeutic Option " the authors studied anticonvulsant potential of andiroba oils (AO1 – acquired from seeds collected in March, and AO2 – acquired from seeds collected in June) and their respective fatty acids amides (FFA1 and FFA2) which were generated by reaction of enzymatic transesterification. The aim of the study is original and well defined. They demonstrated significant anticonvulsant effect of FFA1 in the pentylenetetrazole (PTZ)-induced seizure test in mice. This model is one of the most widely used animal tests in preclinical drug‐screening programs. FFA1 significantly alleviate PTZ-induced myoclonic and generalized tonic-clonic seizures as well as electrocorticographical alterations, i.e., amplitude and energy in Betha rhythm.

            In my opinion, the strong point of the manuscript is precisely arranged analyze of elecrocorticographic activity which is not common method and is infrequently curried out. In contrast, the weakness of the manuscript is using only one of the seizure tests. The authors should have also used other tests (i.e., maximal electroshock or 6 Hz-induced psychomotor seizure test) because using only one test does not give a full picture of the activity of the compound tested. Although FFA1 revealed significant anticonvulsant effect in the PTZ test, it does not have to work in other experimental tests, which might significantly limits its use in epilepsy treatment. Moreover, the effect of flumazenil – an antagonist of GABA A receptors, on anticonvulsant activity of FFA1 was studied only in relation to electrocorticographic activity, its effect was not studied in behavioral test, i.e., the PTZ-induced seizure test in mice. Generally, the manuscript is written in an appropriate way but there are some drawbacks which have to be corrected, i.e.:

1.     The abbreviation FFA should be explained in the “Introduction” part (line 81).

2.     Description of the results should be supplemented by the appropriate statistical tests and their results, i.e. one-way ANOVA with the Tuckey post hoc test, F and p values. These data should be also included in captions for figures.

3.     Some statistically significant differences between the studied groups are mentioned in the “Results” part but they are not marked in figures, i.e. line 123-124: “BDZ presented statistical differences in relation to PTZ and the other evaluated groups” – only difference in relation to PTZ group is marked in the figure; line 151-152: ” The Vehicle + PTZ group presented a mean power of 10.63 ± 4.191 mV2 151 / Hz × 10−3, being statistically different in relation to the other groups” – this difference is not marked in the figure.

4.     Why the data concerning locomotor activity and sedation are not presented in the manuscript. There is no information on how these parameters were evaluated.

5.     Caption in Figure 6 should be improved, because there were 4 groups – control (vehicle treated) and 3 groups which were treated with vehicle, FFA1 and BDZ in combination with PTZ. PTZ is not mentioned in the caption.

6.     Line 173-175, sentence “In the mean power evaluation,…” - there are two values given for the PTZ group, i.e., 0.33 and 10.63 mV2/Hz x 10-3, and in my opinion (according to Fig 7A), only one of them is correct (i.e., 10.63).

7.     Discussion should be improved because it is somewhat chaotic, some information is repeated, i.e. lines 211-214 and lines 264-267. In my opinion, the authors should first discuss the results of the behavioral PTZ test and then the results regarding the electrocoticographic records.

To sum up, I think that the manuscript present novel and interesting results and it might be published after the precise revision.

Reviewer 2 Report

I have a problem with describing flumazenil as an antagonist on GABA-A receptors.  It is what is correctly known as a neutral modulator at these receptors.  It can neutralise the effects of certain positive and negative GABA-A modulators that act on the high affinity benzodiazepine site.  This needs to be corrected throughout the manuscript.

Did flumazenil completely block the anticonvulsant effects of FAA1?  If it did not the anticonvulsant effects may be due to more than one action, it is after all a mixture of substances.

Reviewer 3 Report

This study investigated the anticonvulsant effect of fatty acid amides (FAA) in pentylenetetrazole (PTZ)-evoked the seizures.  These data are interesting, and it seems that FAA1 has some translational potential to control seizures.  I have the following comments for authors:

The authors mentioned the interactions of endocannabinoids with GABAA receptors in multiple places. It was briefly described in the Introduction section that andiroba oil contains substances that can be used as precursors for the synthesis of endocannabinoids.  However, it is not clear what is the relationship of FAAs with endocannabinoids. This should be clarified.  In the Results section, the comparisons among different groups were not appropriately described. For example, the vehicle control groups in figure 2 were referred to as the PTZ groups.  This is confusing, as PTZ was treated in each group.  In the experiment on flumazenil reversal of the anticonvulsant effect of FAA1, a flumazenil control group should be included.

Round 2

Reviewer 1 Report

I my opinion  manuscript entitled “Fatty Acid Amides Synthesized from Andiroba Oil (Carapa guianensis Aublet.) exhibit Anticonvulsant Action with Modulation on GABA-A Receptor in Mice: A Putative Therapeutic Option” has been significantly improved. However, the authors complimented the manuscript with p values of post hoc tests (Tukey test) but there is still no results, i.e., F and p values, of one-way ANOVA. In my opinion, these results are necessary in the manuscript.

Reviewer 3 Report

The authors have addressed all my concerns. 
